# How did communities in North West England respond to the COVID-19 lockdown? Findings from a diary study

Fiona Ward, Emma Halliday, Vivien Holt, Koser Khan, Gill Sadler, Paula Wheeler, Joanna Goldthorpe ⓘD

Applied Research Collaboration North West Coast, Division of Health Research, Lancaster University, Lancaster, UK

**Correspondence to**
Joanna Goldthorpe;
j.goldthorpe@lancaster.ac.uk

## ABSTRACT

**Objectives** During the COVID-19 pandemic, the UK government and public health leaders advocated for community level responses to support vulnerable people. This activity could be planned and co-ordinated, however much was informal and developed organically. The effects on the individuals who were involved in providing and receiving informal support and implications for their communities have not been widely explored. The aim of this study was therefore to document and explore the nature, potential effects and longevity of community responses to the COVID-19 pandemic.

**Participants** We asked 15 individuals in North West England to keep a diary during the first UK COVID-19 lockdown. Over 8 weeks, diaries were completed and supported with weekly calls with researchers. A community capacity building framework was used to explore reported community responses to the COVID-19 pandemic.

**Results** Diarists described community characteristics that enabled and hindered helpful responses in the lockdown context. Diarists frequently described informal approaches with residents acting alone or with near neighbours, although there were examples of community networks and residents recommencing formal volunteering activities. Diarists reported communities providing practical help and social support to vulnerable people. Participants perceived a greater sense of community, increased contact between residents and new networks during the period covered.

**Conclusion** The diaries provided valuable insights and the framework was a useful tool to explore the COVID-19 lockdown context. The findings indicate that organic capacity building took place, primarily via individual agency, highlighting the risk of communities being 'left behind' if there were not individuals or community networks available with resources to plug gaps in organisational support. Recommendations to sustain helpful responses to the pandemic include further consideration of ongoing community mobilisation, empowerment and community control within the capacity building framework.

## STRENGTHS AND LIMITATIONS OF THIS STUDY

⇒ The study's diary methodology allowed for data to be collected in real time during the first COVID-19 pandemic lockdown, capturing authentic, in-the-moment insight from participants.
⇒ The findings were organised around an established framework (Chaskin's framework of community capacity building).
⇒ The findings reflected the experiences of an older, mostly female experience with previous experience of volunteering and/or community service.

reducing associated morbidity and mortality. Despite government and commercial organisations declaring that 'we are all in it together',[1] not everyone experienced the pandemic in the same way due to the effects of national and international structural and systemic inequalities. The effects of the COVID-19 pandemic are being felt particularly severely by clinically vulnerable individuals and globally along racial, ethnic and socioeconomic divides.[2–4] These disproportionate effects highlight and exacerbate the damaging effect of health inequalities that existed before the pandemic and influenced the publication of the Marmot review.[4] This report examines the reasons behind the UK's status as having one of the highest COVID-19 mortality rates in Europe and describes how we can 'build back fairer' by introducing polices that reduce health inequalities.

Research from previous infectious disease pandemics indicates that smaller scale social and community responses may be especially important in reaching vulnerable and marginalised populations. A rapid review of research into community responses to recent pandemics, including Ebola, SARS, Middle East Respiratory Syndrome and H1N1,[5] suggest that they may be crucial in engendering trust and goodwill in those who are suspicious of the motives of national governments and international organisations such

## INTRODUCTION

Reaching and supporting people at risk, particularly those living in disadvantaged communities, has been vital to containing the effects of the COVID-19 pandemic and

as WHO. This trust and credibility is vital for delivering appropriate health messages,[6] modelling behaviours, and establishing social norms that reduce viral transmission.[7]

Public health leaders in the UK proposed and advocated for community responses to support vulnerable people during the pandemic. These included a 'call to arms' for those with basic medical training to deliver frontline home support[8] and a framework for a structured 'whole systems' approaches to community-based initiatives in order to build community resilience (in addition to capacity building around virology, vaccinations and reducing viral transmission).[9] These approaches would have been unlike most identified by Gilmore *et al*,[5] which found community engagement in high-income countries (Australia, Canada and USA) usually entailed passive involvement such as targeted consultation with minority populations and very few had an equity focus or reported informal, locally driven activity.

To date, several quantitative studies have looked at volunteering during the pandemic. They have reported on the demographics of volunteers during the early lockdown period (people who were older, non-white and not living with partner), the categories of volunteering (formal, social action and community)[10] and how they were mobilised (social networks and social media groups).[11] A survey carried out on behalf of the UK Department of Culture, Media & Sport and Office for Civil Society looked specifically at the nature of informal and formal volunteering. It found that participants volunteering informally were more likely to: keep in touch with someone who has difficulties getting out; carrying out tasks such as shopping and collecting medicine and less likely to provide transport or escort to venues outside the home.[12] This research did not, however, explore the experiences of volunteers or others in their communities.

Gilmore *et al*[5] concluded that little attention has been given to informal and unstructured responses to the COVID-19 pandemic carried out by citizens in their own localities and that future pandemic research should include a focus on documenting organic community engagement. This research should additionally be of interest to readers concerned with community empowerment and place-based approaches; citizen-led community initiatives have potential to positively disrupt the power relationships inherent in top-down community development initiatives, resulting in potential benefits for public health.[13] If we are to 'build back better' and 'fairer', it makes sense to capture, support and resource acceptable bottom-up, equitable models of community engagement that have potential to reduce health inequalities.

Chaskin's (2001) definition and framework for community capacity offers a useful lens through which to explore organic community responses to the recent pandemic. It provides a structure for exploration around how local resources were informally mobilised to solve neighbourhood-level problems, and the parameters of these efforts. There may be a point at which individuals and resources come together to build capacity in communities, but also a point at which this capacity is reached and more formal, planned approaches may need to be mobilised. Chaskin[14] conceptualises community capacity as[14]

> …the interaction of human capital, organizational resources, and social capital existing within a given community that can be leveraged to solve collective problems and improve or maintain the well-being of a given community. It may operate through informal social processes and/or organized effort. (18, p.295).

The study was exploratory and sought to broadly understand public experiences of the pandemic including its effects on diarists' everyday lives. This paper uses Chaskin's framework to present perspectives on one of the prominent themes—that of local community responses to the first COVID-19 lockdown. Our specific research question is: How did communities in North West England respond to the COVID-19 lockdown?

## METHODS

### Design

This study used solicited diaries, a research method employed in previous crisis situations to record personal experiences and observations.[15 16] Weekly calls complemented the written diaries as these have been found to aid diary completion and improve the quality of the information recorded.[17]

### Setting

The study was undertaken by researchers from the Applied Research Collaboration North West Coast (ARC-NWC), a collaboration of universities, NHS providers and commissioners, local authorities and third sector organisations which support applied research that responds to the needs of local populations and health and care systems. Public and community involvement is central to this work and the study participants were ARC-NWC public contributors or 'Public Advisers'. All the participants lived in communities based in the North Western coastal area of England.

### Participants

All Public Advisers registered with ARC-NWC were invited, by email, to take part in the study. This means that the participants and researchers may have met previously. A total of 16 people expressed an interest, although one withdrew before the diaries commenced. The remaining 15 completed the 8-week study. All were aged over 18 years. Although not purposively sampled, there was diversity in the circumstances of those who took part. They included people living alone, with families, in clinical at-risk groups, shielding households and people with caring responsibilities. Characteristics of participants are shown in table 1. Written consent was obtained from diarists, who were also reminded of their rights during each weekly call.

**Table 1** Characteristics of participants

| Age | N | Employment status | N | Gender | N |
|---|---|---|---|---|---|
| 30–39 | 1 | Self employed | 2 | Female | 10 |
| 40–49 | 2 | Retired | 4 | Male | 3 |
| 50–59 | 5 | Not employed—carer | 2 | Not stated | 2 |
| 60–69 | 2 | Not employed—other | 5 | | |
| 70 | 3 | Not stated | 2 | | |
| Not stated | 2 | | | | |
| Ethnicity | | Disability | N | | |
| White—British | 9 | Yes | 5 | | |
| Black African and White | 1 | No | 6 | | |
| Pakistani | 3 | Prefer not to say | 2 | | |
| Not stated | 2 | Not stated | 2 | | |

### Patient and public involvement

ARC-NWC has paid public advisers who work alongside researchers at each stage of the research. Each ARC-NWC theme (research team) has a public adviser as a colead to lead on prioritisation and public involvement. Teams involved in specific research studies recruit additional public advisers with lived experience in relevant areas. For this study, public advisers were involved in prioritising, designing the research and checking the analysis, in addition to being participants in the research.

### Procedure

A link to the diary template (created on Qualtrics online survey software) was emailed to the participants (hereafter referred to as diarists) each Monday for 8 weeks from 20 April 2020. A Microsoft Word version was also available. The diary format used a combination of closed questions which enabled diarists to rate their general and emotional health and quality of life each week and free text for them to record, as they chose, their daily activities and general reflections about their current situation. Diarists were not routinely asked specific questions about the pandemic: the exception was in week 7 when they were asked to reflect on major change in lockdown rules.

Each diarist was paired with a researcher (EH, KK, FW and PW) for an induction session and the weekly calls. The conversation during the telephone or video calls, which lasted approximately 30 min, was guided by a proforma which included prompts about general wellbeing and any standout events during the week. Calls were not recorded but notes taken and these were included in the analysis. At the end of the diary period, diarists took part in a focus group which explored their experience of the diary method and participating in the study. To enable diarists to speak openly, one focus group (95 min) was facilitated by researchers not involved in the weekly calls (VH and GS). There were 11 participants.

### Analysis

The data from 115 diary-weeks, fieldnotes from 114 calls and the focus group transcript, were analysed using NVIVO-V.12 to organise the data (EH and VH). An inductive thematic analysis was used to organise the whole data set according to patterns identified by the coders (FW and EH) and to draw out similarities and differences across participants' experiences.[16] One of the prominent themes was the diarists' observations of and role in community responses to the pandemic. To explore this aspect of the data in a more structured and focused way, we conducted a deductive analysis[17] applying Chaskin's framework[18] (FW) and narrative techniques to the analysis of data coded to all relevant themes in our data set (namely community support and responses; social networks and volunteering). To establish trustworthiness of the findings,[19] the research team met to discuss the analysis and diarists participated in a workshop to discuss preliminary findings.[20 21] Quotations used here to illustrate findings are referenced using the diarist's unique ID[1–16] and the diary week.[1–8] Where data come from weekly calls, this is stated (eg, *diarist5-week1-call*).

### RESULTS

The analysis of the experiences and observations of the diarists on the theme of community responses are described below as they relate to the dimensions of the Chaskin framework (figure 1). The framework itself has been reorganised to lead with the 'conditioning influences' as the timing and purpose of the study meant that the lockdown context was the critical circumstance.

### Conditioning influences: the lockdown context

Conditioning influences (dimension 1) are the 'mediating circumstances that may facilitate or inhibit community capacity and efforts to build it' (18, p.299). Although the Diary Study participants lived in different neighbourhoods, there were common factors that encouraged and inhibited the development of community capacity during the first lockdown.

### Residential stability and activity close to home

Diarists described the way that the lockdown resulted in the adoption new routines closer to home. Under

 

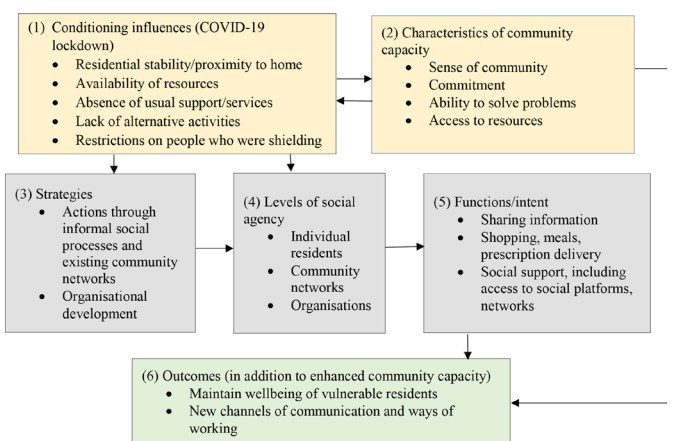

**Figure 1** Chaskin framework dimensions applied to diary project findings.

national rules, people were permitted to exercise daily, and this was seen as an opportunity for social connectivity, particularly for those who lived alone. Several diarists said there were more people walking locally, resulting in additional contacts and casual conversations. As one diarist wrote, 'when I take my daily exercise, people are being more friendly and actually saying hello and even stopping to have a bit of chat' (diarist10-week1). During a catch-up call, diarists said they were 'uplifted' by such greetings and people acknowledging one another. For some, religious occasions which could not be celebrated in the usual way also provided an opportunity for neighbours to connect: one diarist described that during the festival of Shavuot, a neighbour celebrated by sharing food with others on the street (diarist2-week6) and similar activity was reported during Ramadan.

There was, however, less of a sense of community for some because of where they lived. Two diarists who were also shielding described their neighbourhood as consisting of larger houses with older neighbours who rarely saw each other and another lived in a rural location, out of sight of other houses.

### Personal circumstances

Changes in routine, the furlough scheme and the lack of alternative activities created human capacity and facilitated local community engagement, illustrated by one diarist who said that the suspension of their previous volunteering roles had 'led them to find other things to do to support people' (diarist1-week1-call).

Some diarists, however, were not able to undertake volunteering activities that were previously 'a big part' (diarist6-week5-call) of their lives and described the subsequent sense of loss. This was particularly the case for those who were shielding. The change from their normal activities was especially difficult as one diarist wrote:

Inclusion is the biggest value—we are a couple with careers based on helping people be at their best. But now can't be part of this… The sense of being part of a community and not being able to contribute when

you would normally be in that role—it is frustrating—feel a bit jealous not being part of the community coming together (diarist4-week1).

At the end of the study, they were still feeling restricted and a sense of isolation:

I love getting 'stuck into' work and doing things— nearly always for others. I am finding it exceptionally hard not to be able to do this being shielded. I had to call on a local volunteer to get my prescriptions from the chemist and whilst it was wonderful to receive the help it also made me more frustrated that I wasn't the one doing this! (diarist4-week8).

### Community capacity: underlying characteristics

Chaskin describes elements underlying the development of community capacity (dimension 2), including a sense of community, committed individuals, the ability to solve problems and access to resources.

### Opportunities to connect and observe

The Diary Study findings suggest that during the early lockdown there was, in general, a growing sense of connectedness. This was often a physical occurrence, facilitated by socially distanced contact outdoors, but also happening online and via telephone calls. One diarist, for example, reflected on the response when a funeral precession passed along their street saying 'it brought everyone out to pay respects as it passed. It's unusual for the place to be so busy during the day and it showed the sense of community spirit that has emerged': they felt that this indicated 'how much community spirit there was in the locality which people hadn't realised' (diarist1-week1-call).

Several diarists wrote that more frequent conversations between neighbours had resulted in a greater level of understanding about the needs of people at risk in their communities. One diarist, for example, said that when they were delivering food parcels '… many residents are now asking for other help/support/advice, TV licenses, Utility bill payments etc' (diarist11-week2) and another described how a WhatsApp group had 'been a way to identify/support people perceived to be at risk on an informal basis, such as a group member with mental health' (diarist1-week1-call). One diarist described how their father's neighbours had responded during the lockdown:

My dad has always had good neighbours but now they have become fantastic… (they) have joined together to make sure all the old and vulnerable people are supported both emotional and practically. (diarist9-week4)

For several diarists, however, the perceptions of a mutuality of circumstances faded as the lockdown progressed. Towards the end of the study, some expressed frustration at the response of neighbours when restrictions were being lifted, which arguably worked against a sense of community at this point.

## Individuals committed to problem solving

The majority of the diarists, both male and female, were regularly involved in some sort of community activity with a small number engaged almost daily in community responses. During the early weeks of the lockdown, organisations often suspended support services and many people, particularly those who were shielding, were isolated and struggled to obtain essentials such as food shopping. It was clear that despite any personal difficulties' diarists were contending with, many were still thinking about maintaining contact and providing practical help to others in their community. In several cases, they supported people they had met through previous volunteer roles: one 'buddied up' for walks in the park and another took things to someone while they were in hospital and checked their house.

Several diarists described their continued involvement with third sector organisations that they volunteered with and the sense of purpose these activities provided. Many activities had ceased because face-to-face contact was not an option but during the period covered by the diary project, some were reconfigured and moved online. When the pilot call for a support group went well, one diarist described it as 'great news!' When they also received an email asking if they were interested in helping with a new telephone support line they 'signed up straight away and awaited my first client for a phone call on Thursday' (diarist1-week3).

## Access to means and resources

The availability of resources such as telephones and the internet facilitated the development of community capacity. Despite physical restrictions for some, diarists described telephone calls with isolated friends, family and neighbours. Another common theme was the way that diarists adapted, using forms of communication that were new to them, such as Zoom, Microsoft Teams and What's App to maintain existing connections and engage in new activities such as religious services, fundraising, individual and street level support, and volunteering activities.

Diarists also described the additional resources that were made available to support community initiatives including donations from individuals, businesses and social enterprises and in one case, local authority funds. One diarist involved in a community response, for example, described donations of food from local restaurants, meals being prepared in a school kitchen and resources being gathered and packaged in space made available by a community hub (diarist11-week1). Others referred to restaurants and cafés delivering to vulnerable people (diarist10-week1) and community hampers for a local nursing home (diarist14-week1).

## Actors, activities and strategies: delivering the community response

Strategies are the means through which capacity is built or engaged (dimension 3). Chaskin suggests that these often include a combination of community organising, organisational and leadership development or through fostering collective relationships among organisations. Strategies may operate through informal social processes, organised community processes or formal targeted efforts and, to convert these strategies into action, community capacity is engaged through agency stemming from individuals, networks and organisations (dimension 4). The 'intent' of the community capacity building is described as the function (dimension 5).

## Individual acts

Although there was a degree of strategic development, the most frequently reported approaches to community capacity building operated through informal interactions with individual residents proactively applying knowledge, skills and resources. Ten diarists described the instrumental and emotional support they provided (or observed others providing), often to those who were elderly or shielding. Throughout the 8-week study, there were frequent descriptions of 'people helping each other out' (diarist1-week3-call). Activities or functions included practical tasks (eg, shopping, fetching prescriptions and delivery medicines) and providing support through socially distanced walks and telephone calls.

## Informal networks

Community capacity was also developed through networks of relationships with neighbourhood groups and organisations acting together. Several diarists described their membership of social media groups which provided a forum for requests and offers of help from one resident to another. One diarist described how they identified a need for this and set up a street-level WhatsApp group, saying that they were 'enjoying being able to support my street through social media. There is always something to offer from shopping to advice' (diarist 14-week1). Another diarist was heavily involved in a response co-ordinated by a town council. This approach was enabled by local government funding and individuals who were already active in their community mobilised social enterprises, businesses and other local service providers to provide essential shopping and meals to residents at risk. Their local knowledge, ability to act quickly and being known in the community were seen to be essential components underlying their rapid community response (diarist11-week1-call).

## The contribution of organisations

At an organisational level, Chaskin suggested that the development of community capacity 'might be reflected in the ability of such organizations to carry out their functions responsively, effectively, and efficiently as part of the larger system of actors and processes to which they are connected, within and beyond the community' (18, p.298). Positive reporting of organisational community capacity building in the diaries largely focused on organisations adapting support service to be delivered online and there was enthusiasm when, a few weeks

after the commencement of lockdown, diarists were able to engage in volunteering roles. This included one diarist who became very actively involved in an initiative to support people with mental health problems and a befriending project. There was also sympathy expressed for the response of professionals such as general practitioners and the clergy who were seen to be 'reaching out a much as they can but they can't fill the gaps' (diarist4-week8).

But during the initial stages of lockdown, several diarists expressed the view that the needs of vulnerable residents were most frequently being met by individuals or community networks. Diarists expressed concern that organisations needed to find alternative ways to provide for service users reliant on them for social contact and support (eg, diarist1-week2-call) and several suggested that organisations had often been slow to respond to the needs of the community or provide guidance to inform community-based action. Frustration was expressed about what was seen as the lack of effective response from 'well paid people': one diarist described this as 'a disconnect between 'grassroots' organisations and individuals doing stuff, and… councils and others talking about the doing' (diarist11-week1-call) and another wrote about a lack of co-ordination when they did react:

> …there are numerous voluntary organisations that do not like working with each other, they seem to have lost focus as to why they exist. They all share one thing in common, they have definitely forgotten that they exist to serve and support the community and if an individual organisation can't provide that support, then signpost them to one which can (diarist10-Week2).

It was also suggested that in some cases, organisations were unable to use potential community action because of pre-existing organisational arrangements, thus inhibiting the potential impact of more informal groups. One diarist reported that statutory funding was not available to community groups who were ready to respond to local need because they were not formally constituted. This was at a time when some organisations which could have been part of the community network had furloughed staff, so restricting the availability of people in key roles.

### Outcomes of the increased community capacity

The outcome of the community activity (dimension 6) as identified and described by the diarists included an increase in community capacity (dimension 2) as well as outcomes relating to the tasks undertaken.

### Maintaining well-being

Maintaining the well-being of residents at risk was highlighted as a motivation for the community responses. This support was clearly valued as one diarist wrote:

> People are mentioning that they will really miss us tomorrow!!! It's a very humbling feeling, knowing that

a small meal, a chat and a smile can mean so much to someone… (diarist11-week3)

Several diarists also described the importance to their personal well-being of being engaged in community or volunteering work at such a challenging time. They described these activities as 'giving a sense of purpose' (diarist3-week2) and enhancing feeling of self-worth through being 'useful' 'of value' and 'productive' (diarist11-week1). Another diarist described the mutual benefit of being involved in a new support initiative:

> Had my first Check in and chat call today, went brilliantly on the phone for an hour and I think we both got something out of the experience. Look forward to my next chat next week (diarist1-week3).

But this activity also had personal physical and mental consequences. One diarist was heavily involved in the mobilisation of community network which collected, processed and delivered food to vulnerable residents described how demanding it could be:

> Over at (locality2) primary school to collect pasta/ soups to package and distribute later today. 11.00am, 120 food/sandwiches packages ready to load onto van, the list is growing, some residents ask to be removed as they have adequate support networks, but for every 1 we take off, 4 more ask for support. Finished delivering food around 5.45pm, really tired (diarist11-week2).

In addition to being active and engaged in proving support to others, diarists were likewise experiencing the pandemic and were dealing with the psychological impact of this. This diarist also noted the additional strain associated with being out in the community and having contact with vulnerable people:

> Just been told that today, will be our final day of delivering support as part of the (organisation) COVID-19 community response. Such a mixture of feelings. Relieved, and happy that I (along with 10 of my friends and colleagues) have so far survived contracting the virus, despite placing ourselves (and our loved ones) in danger. The stress of being in close proximity with vulnerable residents, many of them elderly, with multiple and complex health conditions has been enormous at times (diarist11-week7).

### New connections and information sharing

The overall perception from the diarists was that the increased contact between residents had resulted in the development of new networks and channels of communication. In turn, this had led to a greater awareness of the needs of individuals within the community, information sharing and pathways for individuals and businesses to 'help out more'. Diarists also described, through their own volunteering experiences, the new initiatives and ways of working developed by the organisations they were

connected with to meet the support needs of their service users.

## Longevity of the community capacity

The nature of the Diary Study, in gathering the perceptions of 15 individuals during the early weeks of the lockdown, meant it was not possible to comment on the extent to which the community capacity built or other outcomes were sustained beyond the diary period. Even during the period covered by the Diary Study, activities were changing and in one instance, a community response that had regularly delivered shopping and meals to more than 100 people was being 'wound down'. In part, this was an apparent reaction to fears that they were 'creating dependency' but also with the understanding that furloughed staff were returning to work and there was some return to normality.

But several diarists, as illustrated below, did express their hope that the greater sense of community and capacity built would be sustained:

Making sure we don't lose the positives There have been so many positive things that have come from the Covid situation. So much connectivity between people, huge numbers offering help, old friendships renewed, new friends being made, lots of creative solutions being identified to new and difficult problems. I really hope a lot of these really beneficial things can be captured and encouraged to still remain in place when life gets a bit more back to normal (diarist4-week8).

There was a hope that this enthusiasm could be captured but clearly also some doubt—with reference to the community connectedness they had observed, this diarist said 'What a shame we can't bottle it!' (diary4-week3).

## CONCLUSIONS

The research reported here aimed to explore community responses to the COVID-19 pandemic. While the data from this Diary Study focused on emerging community responses rather than planned initiatives, Chaskin's exploration of community capacity proved to be a useful framework to understand the observations and experiences of the diarists during the early weeks of the first UK COVID-19 lockdown. Though actions were largely reactive, the findings did indicate that capacity building took place: participants described a sense of community, and a desire to be actively engaged; the generation of ideas and solutions to local issues as they were identified and harnessing of resources were apparent in data from diary entries and weekly calls.

Using Chaskin's framework also highlights the importance of conditioning influences and in the predicament created by the COVID-19 lockdown, the accounts of the diarists elucidating the critical role of context in the development of community capacity. Inhibiting influences include the lockdown rules, restrictions on people

who were shielding and the absence of existing services and other sources of support. The diarists described how community capacity building was supported by people's physical proximity to home, the use of information technology, greater availability and by goodwill which frequently appeared to engender a feeling of being 'in it together'.

The sample included in this study was predominantly female. The social and economic effects of the pandemic are likely to have been experienced differently for men and women. Research suggests that the burden of unpaid care work already disproportionately carried out by women was exacerbated by the pandemic in ways that have the potential to be detrimental to their well-being.[18 19] However, data from this study suggest that prepandemic volunteering experience and connections supported the participation of both male and female diarists in community responses during the pandemic. The extensive involvement reported by male diarists in community capacity building activities contrasts with other research which highlights the disproportionate contribution of women as voluntary care givers during the pandemic.[19] The findings presented may therefore be representative of people who were already familiar with carrying out unpaid caring and supporting roles.

The picture of the community response presented by the diaries contains insights into different population characteristics that could inform the recovery. It supports research which suggests that residents must be engaged if more systemic recovery support is to be valued, for example, reference[20] and reveals valuable information from members of the community about the knowledge and processes that resulted in the provision of essential support while there were gaps in service delivery. As the National Institute for Health and Care Excellence has indicated, the recovery period provides an opportunity for local government and health and care services to build on the strengths of communities to ensure initiatives are 'relevant and meaningful'.[21]

The main functions of the community response described by the diarists was to provide essential resources to residents at risk, and to a lesser extent, social support. The sharing of intelligence was also required to enable this to happen, a finding in line with other work which described the 'creative solutions' developed by residents who understand their community.[22] While diarists wrote about the communication of lockdown rules and government health messages, there was no evidence that this formed any part of their activities, contrasting with the type of activities described in previous epidemics.[7] This finding supports the UK Government's research that found informal volunteers were most likely to be employed maintaining social contact with less mobile neighbours and carrying out practical tasks such as shopping and collecting medicine.[12]

Support was provided through community organising, primarily via individual agency and local connections and, to a lesser extent, organised community processes.

This resonates with other research findings[5] where the main actors in community pandemic responses were community leaders, community and faith-based organisations, community networks, committees and individuals. Some diarists referred to targeted efforts of larger organisations but in general, the activities described were not part of a strategic approach, rather they were informal and evolving in response to the initial crisis and rapidly changing circumstances. The prominence of these individual and self-organised street-level activities appeared to be effective for those residents who were identified: practical support was provided at a time when formal support services were not available or not able to cope. Further supporting the findings of Gilmore *et al*,[5] there was no discussion in the diaries about community involvement in wider strategic decision-making.

The greater community capacity generated during the time covered by the Diary Study appears to have been created through a wider awareness of need within the community and the development of channels to share this information and respond to need (eg, through social media groups). Other outcomes include access to new sources of support such as cowalking and befriending which enabled the well-being of residents at risk to be maintained. It is also important to acknowledge the personal toll for those involved in a community response: volunteers were similarly experiencing the lockdown with its associated strain on themselves, their families and acquaintances in addition to experiencing anxieties about their contact with infected and clinically vulnerable people. This raises questions about appropriate support and the resources required to sustain successful elements of community-based responses during the COVID-19 recovery and beyond.

In contrast with other community action frameworks, the concepts of empowerment and control do not feature prominently in Chaskin's dimensions of community capacity building. Similarly, the processes and outcomes described by the diarists did not include references to feelings of empowerment and control. One apparent reason for this may be that the unique context generated by the COVID-19 lockdown meant that the singular concern of many individuals, networks and organisations was to meet the immediate needs of residents most at risk. Consequently, in this context, Chaskin's framework was useful to describe this 'inward gaze'[13] with its focus on, for example, community capacities, and neighbourhood conditions. But moving forward, and beyond the COVID-19 recovery, it is argued that power and control need to be integral elements in the consideration of community capacity, requiring an 'outward gaze' on the political and social change required to enable structural gaps and inequalities to be addressed.

The risk of community capacity building being considered out with empowerment and control is a particular concern at a time when there are reductions in state spending and, as Popay *et al*[13] argue, pressure to focus on 'equipping communities to use their "assets" to manage "shocks" like COVID-19—to adapt to, rather than transform, existing inequalities' (p.8). With reduced public expenditure, process of 'citizen-shift' has also been described whereby interventions are increasingly individualised and service providers are enabled to create upstream energy from grassroots level to mobilise against inequalities.[23] This is an important area for future research and could inform strategies to 'build back better'.

Some potential limitations of the study should be noted. First, the small participant sample (n=15), largely reflected an older and predominantly female demographic and people, as Public Advisers, who were already connected into community and volunteering networks. Although not necessarily a limitation, the findings may therefore not be reflective of the general population. Second, although the findings of this study suggest that community capacity building took place, the short-term nature of the study means that it is not possible to assess the extent to which it was sustained or whether there was a lasting impact. Finally, the diary entries and calls also do not provide a complete picture of activity with local communities, rather they are the observations and experiences these individuals chose to report: the method adopted meant the agenda was set by diarists and researchers did not ask diarists about any possible 'gaps' in their accounts.

There have been few qualitative diary studies completed during crisis situations[5 10–12] and the findings of this research suggest that the combination of written diaries and weekly calls was a useful way to gather the experiences and observations of participants during the COVID-19 lockdown. The study also contributes to building wider knowledge around community responses in the COVID-19 pandemic.

The findings illuminate the community responses experienced and observed at a time of considerable uncertainty and threats to well-being resulting from the COVID-19 pandemic. Chaskin's community capacity building framework focused analytical attention on the conditional influences and underlying characteristics of community capacity in addition to the strategies, actors, intent and outcomes. This demonstrated the complexity of the situation and the central importance of the specific context and of individual agency: residents were engaged (sometimes intensively) in developing links, solutions, working with what they had and, in some cases, with organisations that were in a position to respond. This confirms the likelihood that many communities could be left behind if there were not equipped individuals and/or community networks able to fill the gaps in the organisational response, at least in the short-term. This could lead to increased health inequalities.

A number of directions for future research have emerged from the study. These include insights from residents who began their volunteering journey during the pandemic and looking at the impact of the community response on the well-being of those involved as well as those being supported. Investigating whether individuals and community networks involved in the immediate response were mobilised in the recovery postpandemic would be valuable: embedding an 'outward gaze' into future research could

also ascertain the extent to which community empowerment and involvement in wider decision-making results in strategic dimension to community capacity building post-COVID-19.[5] Identifying commonalities or differences in experiences across communities could also provide valuable information about inhibiting influences for 'communities left behind. In addition, this could also help us to explore relationships and networks that have been the most powerful in responding to local needs.

**Acknowledgements** The authors wish to thank all ARC-NWC Public Advisers who participated in the study and to Alan Price and Neil Joseph who reviewed the paper. Thanks also to Olga Angulo-Judez, Lancaster University, who provided technical support and training for diarists in the use of online methods. This paper is dedicated to the memory of Gill Sadler who sadly passed away in January 2021.

**Contributors** FW (guarantor) designed the study, developed the protocol, co-ordinated the data collection, analysed the data and wrote the manuscript; EH designed the study, collected and analysed the data and reviewed the manuscript. VH collected the data and reviewed the manuscript; KK collected the data and reviewed the manuscript; GS collected the data; PW collected the data and reviewed the manuscript; JG wrote the manuscript and provided methodological and analytical support.

**Funding** This research was funded by the National Institute for Health Research (NIHR) Applied Research Collaboration North West Coast (ARC-NWC).

**Competing interests** None declared.

**Patient and public involvement** Patients and/or the public were involved in the design, or conduct, or reporting or dissemination plans of this research. Refer to the Methods section for further details.

**Patient consent for publication** Not required.

**Ethics approval** This study involves human participants and was approved by Lancaster University Faculty of Health and Medicine Ethics Committee April 2020 (FHMREC 19076). Participants gave informed consent to participate in the study before taking part.

**Provenance and peer review** Not commissioned; externally peer reviewed.

**Data availability statement** Data are available upon reasonable request. Data available on request due to privacy/ethical restrictions The data that support the findings of this study are available on request from the corresponding author (JG). The data are not publicly available due to their containing information that could compromise the privacy of research participants.

**ORCID iD**
Joanna Goldthorpe http://orcid.org/0000-0001-7839-7544

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
