## [Reviewer comments · BMJ Open]

ARTICLE DETAILS

TITLE (PROVISIONAL)	How did communities in North West England respond to the COVID-19 lockdown? Findings from a diary study
AUTHORS	Ward, Fiona; Halliday, Emma; Holt, Vivien; Khan, Koser; Sadler, Gill; Wheeler, Paula; Goldthorpe, Joanna

VERSION 1 – REVIEW

REVIEWER	Hoverd , William Massey University, People Environment and Planning
REVIEW RETURNED	17-Nov-2021

GENERAL COMMENTS	Dear Authors, thank you for the opportunity to read your work and I appreciate reading about your study and admire the depth offered by your qualitative method and your innovation in data collection under lockdown. I feel that you have outlined the strengths, limitations and challenges with your method and dataset well. I have three things that I would like you to consider with a revision. 1. What are the overall findings of the study and why should they be of interest to the readers of the BMJ? This needs to be clearer specifically in the abstract and introduction but throughout as well.2. When you introduce Chaskin's model, you use a table without explanation of/for the usefulness and applicability of the model. It looks somewhat arbitrary, could you locate the purpose and function of the model when it is first introduced.3. When one reads the results, one sees that the findings are mostly positive and they are clearly a small amount of the overall data set. It almost feels like they have been selected to fulfil the chosen analytical model. Does the remainder of the data fit? Its not clear. Some scoping and tightening of the data analyzed here would make the results more persuasive.
--

REVIEWER	Ntontis, Evangelos The Open University, School of Psychology and Counselling
REVIEW RETURNED	07-Feb-2022

GENERAL COMMENTS	I had the pleasure of reading the manuscript titled "Community responses to the UK COVID-19 lockdown: Findings from a diary study". The authors report an insightful exploration of volunteers participating in mutual aid and community support using diary data collected over 8 weeks, which complements previous research that to my knowledge mostly uses either quantitative survey data or cross-sectional interviews. Below I will describe some changes that, to my opinion, would further improve the paper.
---

	1) In the method section (and the analysis subsection) it would be useful to describe Chaskin's framework a bit more, as at the moment you only describe Chaskin's definition of community capacity. What is this framework? What are its core positions or main parameters? Why is it useful for this analysis? How does it complement the inductive analysis? Importantly, why is there a need to rework your inductive thematic analysis based on another framework from a deductive perspective, rather than, for example, reporting only the inductive thematic analysis? In general, it would be useful for readers to have some more information on Chaskin's model as well as to better understand why two seemingly antithetical steps were followed for the analysis (first an inductive one followed by a deductive one). 2) You start your results section by stating that "the dimensions of the definitional framework have been reorganised to lead with the conditioning influences". However, these dimensions and their connection to the project were not reported earlier and thus it becomes hard for the readers to follow the analysis and the rationale behind it and its structure. 3) As it stands, the structure of the analysis and the way it links to the framework is not immediately clear to me, especially when considering the use of headings ("COVID-19 lockdown: the overriding context") and subheadings ("Perceptions of community and facilitators of community capacity"). For example, why are Perceptions of community and facilitators of community capacity a subheading of context and not a core part of the analysis? The same point applies to the other sections. In general, apart from minor re-editing of the structure of the analysis, the paper would benefit by a few sentences at the beginning of the "results" section in which the authors outline the structure of the analysis. This will help readers to follow the analysis and to know what to expect. 4) I like the results section as it is clear and precise. One thing that can be improved is the naming of the themes. For example, the title "Perceptions of community and facilitators of community capacity" is not clear enough as it does not indicate precisely what is important or distinct about this theme. Theme titles should capture the key content of each theme in a few words, giving the reader an idea of what they can expect to read. For example, the theme that is currently labelled as "Perceptions of community and facilitators of community capacity" could be somehow renamed so as to describe that the specific facilitators are. One example would be something along these lines: "Facilitators of community capacity: Perceptions of community, opportunities to socialize, presence of mutual aid and social support". This suggestion is not definitive, the authors can find the best way to describe the theme in a few words. Same goes for the next section on inhibitors. Which are these? Please reflect them in the theme name. 5) In the discussion, the authors say that "There have been few qualitative diary studies completed during crisis situations". Could you please provide a few references as examples? 6) Please be clear regarding the institution that provided ethical approval for the study as well as how you recorded participants' consent.
--	--

VERSION 1 – AUTHOR RESPONSE

Reviewer 1 Comment	Response
1. What are the overall findings of the study and why should they be of interest to the readers of the BMJ? This needs to be clearer specifically in the abstract and introduction but throughout as well.	Thank you for your helpful suggestion. We have strengthened our rationale for conducting the research and made this more explicit for the reader. Please see: Abstract, lines 3-9: During the COVID-19 pandemic the UK government and public health leaders advocated for community level responses to support vulnerable people. This activity could be planned and co-ordinated, however much was informal and developed organically. The effects on the individuals who were involved in providing and receiving informal support and implications for their communities has not been widely explored. The aim of this study was therefore to document and explore the nature, potential effects and longevity of community responses to the COVID-19 pandemic. Introduction lines 87-93: Gilmore et al.[5] concluded that little attention has been given to informal and unstructured responses to the COVID-19 pandemic carried out by citizens in their own localities and that future pandemic research should include a focus on documenting organic community engagement This research should additionally be of interest to readers concerned with community empowerment and place-based approaches; citizen-led community initiatives have potential to positively disrupt the power relationships inherent in top-down community development initiatives, resulting in potential benefits for public health Conclusion line 567-570: Identifying commonalities or differences in experiences across communities could also provide valuable information about inhibiting influences for ‘communities left behind. In addition, this could also help us to explore relationships and networks that have been the most powerful in responding to local needs.
2. When you introduce Chaskin's model, you use a table without explanation of/for the usefulness and applicability of the model. It looks somewhat arbitrary, could you locate the purpose and function of the model when it is first introduced.	We agree that Chaskin’s model and rationale for applying it could be introduced sooner in the manuscript. We have therefore added the following text to the introduction: Chaskin’s (2001) definition and framework for community capacity offers a useful lens through which to explore organic community responses to the recent pandemic. It

	provides a structure for exploration around how local resources were informally mobilised to solve neighbourhood-level problems, and the parameters of these efforts. There may be a point at which individuals and resources come together to build capacity in communities, but also a point at which this capacity is reached and more formal, planned approaches may need to be mobilised. Chaskin (2001) conceptualises community capacity as [18] “...the interaction of human capital, organizational resources, and social capital existing within a given community that can be leveraged to solve collective problems and improve or maintain the well-being of a given community. It may operate through informal social processes and/or organized effort.” (18, p.295). The study was exploratory (meaning we did not formulate precise research questions in advance) but sought to broadly understand public experiences of the pandemic including its effects on diarists' everyday lives. This paper uses Chaskin's Framework to present perspectives on one of the prominent themes - that of local community responses to the first COVID-19 lockdown, (lines 99-118)
3. When one reads the results, one sees that the findings are mostly positive and they are clearly a small amount of the overall data set. It almost feels like they have been selected to fulfil the chosen analytical model. Does the remainder of the data fit? Its not clear. Some scoping and tightening of the data analyzed here would make the results more persuasive.	We agree this is important to clarify. We have added a sentence in the introduction (above) and the analysis (extract below in this cell) to make clear this paper covers a subset of the data and that following the identification of community responses as an important theme, the data in all relevant nodes was included. We have also said that the preliminary findings were discussed within the whole research team and with the diarists who participated in the study to ensure they were trustworthy. The data from 115 diary-weeks, fieldnotes from 114 calls and the focus group transcript were analysed using NVIVO-12 to organise the data. (EH, VH). An inductive thematic analysis was used to organise the whole dataset according to patterns identified by the coders (FW, EH) and to draw out similarities and differences across participants' experiences[16]. One of the prominent themes was the diarists' observations of and role in community responses to the pandemic. To explore this aspect of the data in a more structured and focused way we conducted a deductive analysis[17]

	applying Chaskin's framework[18] (FW) and narrative techniques to the analysis of data coded to all relevant themes in our dataset (namely community support and responses; social networks; and volunteering). To establish trustworthiness of the findings[19], the research team met to discuss the analysis and diarists participated in a workshop to discuss preliminary findings[201-213].
Reviewer 2 Comment	Response
1) In the method section (and the analysis subsection) it would be useful to describe Chaskin's framework a bit more, as at the moment you only describe Chaskin's definition of community capacity. What is this framework? What are its core positions or main parameters? Why is it useful for this analysis? How does it complement the inductive analysis? Importantly, why is there a need to rework your inductive thematic analysis based on another framework from a deductive perspective, rather than, for example, reporting only the inductive thematic analysis? In general, it would be useful for readers to have some more information on Chaskin's model as well as to better understand why two seemingly antithetical steps were followed for the analysis (first an inductive one followed by a deductive one).	We agree that Chaskin's model and rationale for applying it could be introduced sooner in the manuscript. We have therefore added the following text to the introduction: Chaskin's (2001) definition and framework for community capacity offers a useful lens through which to explore organic community responses to the recent pandemic. It provides a structure for exploration around how local resources were informally mobilised to solve neighbourhood-level problems, and the parameters of these efforts. There may be a point at which individuals and resources come together to build capacity in communities, but also a point at which this capacity is reached and more formal, planned approaches may need to be mobilised. Chaskin (2001) conceptualises community capacity as [18] "...the interaction of human capital, organizational resources, and social capital existing within a given community that can be leveraged to solve collective problems and improve or maintain the well-being of a given community. It may operate through informal social processes and/or organized effort." (18, p.295). The overarching aim of this study was therefore to use Chaskin's framework to explore the experience of adults living in North West England, including perspectives on their local community's responses during the COVID-19 lockdown (lines 99-115)
2) You start your results section by stating that "the dimensions of the definitional framework have been reorganised to lead with the conditioning influences". However, these dimensions and their connection to the project were not reported earlier and thus it becomes hard for the readers to follow the analysis and the rationale behind it and its structure.	We hope that our response to the point above has gone some way to addressing this helpful comment. We have further added the following to the methods section to improve clarity of our approach: A large amount of data focused on diarists' roles in community responses to the pandemic. In order to explore this aspect of our findings in a more structured and focused way we conducted a deductive, framework analysis[17] using Chaskin's framework[18] (FW)

	and the use of narrative techniques to investigate diarists' experience, including their observations and personal involvement, of community responses over time (lines 177-182)
3) As it stands, the structure of the analysis and the way it links to the framework is not immediately clear to me, especially when considering the use of headings ("COVID-19 lockdown: the overriding context") and subheadings ("Perceptions of community and facilitators of community capacity"). For example, why are Perceptions of community and facilitators of community capacity a subheading of context and not a core part of the analysis? The same point applies to the other sections. In general, apart from minor re-editing of the structure of the analysis, the paper would benefit by a few sentences at the beginning of the "results" section in which the authors outline the structure of the analysis. This will help readers to follow the analysis and to know what to expect. 4) I like the results section as it is clear and precise. One thing that can be improved is the naming of the themes. For example, the title "Perceptions of community and facilitators of community capacity" is not clear enough as it does not indicate precisely what is important or distinct about this theme. Theme titles should capture the key content of each theme in a few words, giving the reader an idea of what they can expect to read. For example, the theme that is currently labelled as "Perceptions of community and facilitators of community capacity" could be somehow renamed so as to describe that the specific facilitators are. One example would be something along these lines: "Facilitators of community capacity: Perceptions of community, opportunities to socialize, presence of mutual aid and social support". This suggestion is not definitive, the authors can find the best way to describe the theme in a few words. Same goes for the next section on inhibitors. Which are these? Please reflect them in the theme name.	We have addressed point 3 & 4 together: This was useful feedback and we agree that the structure of the results section could be clearer. We have added an introductory sentence, re-arranged the material in this section throughout and used the dimensions of the Chaskin framework as the main headings. Sub-headings have also been added so that each theme is more clearly named and reflects the content of the text that follows.
5) In the discussion, the authors say that "There have been few qualitative diary studies completed during crisis situations". Could you	Thank you for pointing out this oversight. References have now been included to support this text (Line 558).

please provide a few references as examples?	
6) Please be clear regarding the institution that provided ethical approval for the study as well as how you recorded participants' consent.	Please see lines 141 and 596-597: Written consent was obtained from diarists, who were also reminded of their rights during each weekly call.

VERSION 2 – REVIEW

REVIEWER	Hoverd , William Massey University, People Environment and Planning
REVIEW RETURNED	22-Mar-2022

GENERAL COMMENTS	Dear Authors, thank you for the opportunity to read through your manuscript. Following through the BMJ criteria, I want to request two minor clarifications within the manuscript. 1. The BMJ criteria ask for a clear research question that the research sets out to answer. While I can see an area for exploration signaled at the outset of the manuscript, I can't see a clear research question to be answered by your study. Rather you explore an area. Some minor reframing here towards answering a specific question would met the journal review criteria. 2. I note that you discuss some limitations of your participant pool in your conclusion. At no point do you discuss gender (your participants are almost all female). Women undertakes much more unpaid labour than men. As such, the fit to Chaskin's model needs to be a little less generalised and more specific/attentive to women's community participation. I would like you be attentive to that dimension of the research. Again, this requires probably a reference or two and a couple of direct sentences.
---

REVIEWER	Ntontis, Evangelos The Open University, School of Psychology and Counselling
REVIEW RETURNED	04-Mar-2022

GENERAL COMMENTS	I have now had the chance to read the revised manuscript. I would like to thank the authors for their work and for seriously considering the comments raised. The manuscript has improved considerably and I have no further issues for the authors to address.
---

VERSION 2 – AUTHOR RESPONSE

1. The BMJ criteria ask for a clear research question that the research sets out to answer. While I can see an area for exploration signalled at the outset of the manuscript, I can't see a clear research question to be answered by your study. Rather you explore an area. Some minor reframing here towards answering a specific question would met the journal review criteria.

Response: Thank you for pointing this out. To comply with the BMJO's requirements we have amended the title to: "How did communities in North West England respond to the COVID-19 lockdown? Findings from a diary study" and added the following text:

The study was exploratory and sought to broadly understand public experiences of the pandemic including its effects on diarists' everyday lives. This paper uses Chaskin's Framework to present perspectives on one of the prominent themes - that of local community responses to the first COVID-19 lockdown. Our specific research question is: How did communities in North West England respond to the COVID-19 lockdown? (108-113)

2. I note that you discuss some limitations of your participant pool in your conclusion. At no point do you discuss gender (your participants are almost all female). Women undertakes much more unpaid labour than men. As such, the fit to Chaskin's model needs to be a little less generalised and more specific/attentive to women's community participation. I would like you be attentive to that dimension of the research. Again, this requires probably a reference or two and a couple of direct sentences.

Response: Thank you for encouraging us to consider this oversight. We have added the following text which we agree signposts to an important context relevant to the findings: The sample included in this study was predominantly female. The social and economic effects of the pandemic are likely to have been experienced differently for men and women. Research suggests that the burden of unpaid care work already disproportionately carried out by women was exacerbated by the pandemic in ways that have the potential to be detrimental to their wellbeing (18, 19). However, data from this study suggests that pre-pandemic volunteering experience and connections supported the participation of both male and female diarists in community responses during the pandemic. The extensive involvement reported by male diarists in community capacity building activities contrasts with other research which highlights the disproportionate contribution of women as voluntary care givers during the pandemic (19). The findings presented may therefore be representative of people who were already familiar with carrying out unpaid caring and supporting roles. (488-498)

Firstly, the small participant sample (n=15), largely reflected an older and predominantly female demographic and people, as Public Advisers, who were already connected into community and volunteering networks. Although not necessarily a limitation, the findings may therefore not be reflective of the general population. (568-572)

This is supported by the following references:

18. Andersen D, Toubøl J, Kirkegaard S, Bang Carlsen H. Imposed volunteering: Gender and caring responsibilities during the COVID-19 lockdown. *The Sociological Review*. 2022;70(1):39-56.

19. Women more likely than men to have tried to help others amid COVID-19 outbreak [press release]. 2020. <https://www.ipsos.com/en-uk/covid-19-coronavirus-outbreak-crisis-polls-community-support-help-women-men> (accessed 29 March 2022)